# Policy Evaluation Using the $\Omega$-Return

**Philip S. Thomas**
University of Massachusetts Amherst
Carnegie Mellon University

**Scott Niekum**
University of Texas at Austin

**Georgios Theocharous**
Adobe Research

**George Konidaris**
Duke University

## Abstract

We propose the $\Omega$-return as an alternative to the $\lambda$-return currently used by the TD($\lambda$) family of algorithms. The benefit of the $\Omega$-return is that it accounts for the correlation of different length returns. Because it is difficult to compute exactly, we suggest one way of approximating the $\Omega$-return. We provide empirical studies that suggest that it is superior to the $\lambda$-return and $\gamma$-return for a variety of problems.

## 1 Introduction

Most *reinforcement learning* (RL) algorithms learn a *value function*—a function that estimates the expected return obtained by following a given policy from a given state. Efficient algorithms for estimating the value function have therefore been a primary focus of RL research. The most widely used family of RL algorithms, the TD($\lambda$) family [1], forms an estimate of return (called the $\lambda$-return) that blends low-variance but biased temporal difference return estimates with high-variance but unbiased Monte Carlo return estimates, using a parameter $\lambda \in [0, 1]$. While several different algorithms exist within the TD($\lambda$) family—the original linear-time algorithm [1], least-squares formulations [2], and methods for adapting $\lambda$ [3], among others—the $\lambda$-return formulation has remained unchanged since its introduction in 1988 [1].

Recently Konidaris et al. [4] proposed the $\gamma$-return as an alternative to the $\lambda$-return, which uses a more accurate model of how the variance of a return increases with its length. However, both the $\gamma$ and $\lambda$-returns fail to account for the correlation of returns of different lengths, instead treating them as statistically independent. We propose the $\Omega$-return, which uses well-studied statistical techniques to directly account for the correlation of returns of different lengths. However, unlike the $\lambda$ and $\gamma$-returns, the $\Omega$-return is not simple to compute, and often can only be approximated. We propose a method for approximating the $\Omega$-return, and show that it outperforms the $\lambda$ and $\gamma$-returns on a range of off-policy evaluation problems.

## 2 Complex Backups

Estimates of return lie at the heart of value-function based RL algorithms: an estimate, $\hat{V}^\pi$, of the value function, $V^\pi$, estimates return from each state, and the learning process aims to reduce the error between estimated and observed returns. For brevity we suppress the dependencies of $V^\pi$ and $\hat{V}^\pi$ on $\pi$ and write $V$ and $\hat{V}$. *Temporal difference* (TD) algorithms use an estimate of the return obtained by taking a single transition in the *Markov decision process* (MDP) [5] and then estimating the remaining return using the estimate of the value function:

$$R^{\text{TD}}_{s_t} = r_t + \gamma \hat{V}(s_{t+1}),$$

where $R_{s_t}^{\text{TD}}$ is the return estimate from state $s_t$, $r_t$ is the reward for going from $s_t$ to $s_{t+1}$ via action $a_t$, and $\gamma \in [0, 1]$ is a discount parameter. Monte Carlo algorithms (for episodic tasks) do not use intermediate estimates but instead use the full return,

$$R_{s_t}^{\text{MC}} = \sum_{i=0}^{L-1} \gamma^i r_{t+i},$$

for an episode $L$ transitions in length after time $t$ (we assume that $L$ is finite). These two types of return estimates can be considered instances of the more general notion of an $n$-step return,

$$R_{s_t}^{(n)} = \left( \sum_{i=0}^{n-1} \gamma^i r_{t+i} \right) + \gamma^n \hat{V}(s_{t+n}),$$

for $n \geq 1$. Here, $n$ transitions are observed from the MDP and the remaining portion of return is estimated using the estimate of the value function. Since $s_{t+L}$ is a state that occurs after the end of an episode, we assume that $\hat{V}(s_{t+L}) = 0$, always.

A *complex return* is a weighted average of the $1, \ldots, L$ step returns:

$$R_{s_t}^{\dagger} = \sum_{n=1}^{L} w_{\dagger}(n, L) R_{s_t}^{(n)}, \tag{1}$$

where $w_{\dagger}(n, L)$ are weights and $\dagger \in \{\lambda, \gamma, \Omega\}$ will be used to specify the weighting schemes of different approaches. The question that this paper proposes an answer to is: what weighting scheme will produce the best estimates of the true expected return?

The $\lambda$-return, $R_{s_t}^{\lambda}$, is the weighting scheme that is used by the entire family of TD($\lambda$) algorithms [5]. It uses a parameter $\lambda \in [0, 1]$ that determines how the weight given to a return decreases as the length of the return increases:

$$w_{\lambda}(n, L) = \begin{cases} (1 - \lambda)\lambda^{n-1} & \text{if } n < L \\ 1 - \sum_{i=1}^{n-1} w_{\lambda}(i) & \text{if } n = L. \end{cases}$$

When $\lambda = 0$, $R_{s_t}^{\lambda} = R_{s_t}^{\text{TD}}$, which has low variance but high bias. When $\lambda = 1$, $R_{s_t}^{\lambda} = R_{s_t}^{\text{MC}}$, which has high variance but is unbiased. Intermediate values of $\lambda$ blend the high-bias but low-variance estimates from short returns with the low-bias but high-variance estimates from the longer returns.

The success of the $\lambda$-return is largely due to its simplicity—TD($\lambda$) using linear function approximation has per-time-step time complexity linear in the number of features. However, this efficiency comes at a cost: the $\lambda$-return is not founded on a principled statistical derivation.[1] Konidaris et al. [4] remedied this recently by showing that the $\lambda$-return is the maximum likelihood estimator of $V(s_t)$ given three assumptions. Specifically, $R_{s_t}^{\lambda} \in \arg\max_{x \in \mathbb{R}} \Pr(R_{s_t}^{(1)}, R_{s_t}^{(2)}, \ldots, R_{s_t}^{(L)} | V(s_t) = x)$ if

**Assumption 1** (Independence). $R_{s_t}^{(1)}, \ldots, R_{s_t}^{(L)}$ *are independent random variables,*

**Assumption 2** (Unbiased Normal Estimators). $R_{s_t}^{(n)}$ *is normally distributed with mean* $\mathbf{E}[R_{s_t}^{(n)}] = V(s_t)$ *for all* $n$.

**Assumption 3** (Geometric Variance). $\text{Var}(R_{s_t}^{(n)}) \propto 1/\lambda^n$.

Although this result provides a theoretical foundation for the $\lambda$-return, it is based on three typically false assumptions: the returns are highly correlated, only the Monte Carlo return is unbiased, and the variance of the $n$-step returns from each state do not usually increase geometrically. This suggests three areas where the $\lambda$-return might be improved—it could be modified to better account for the correlation of returns, the bias of the different returns, and the true form of $\text{Var}(R_{s_t}^{(n)})$.

The $\gamma$-return uses an approximate formula for the variance of an $n$-step return in place of Assumption 3. This allows the $\gamma$-return to better account for how the variance of returns increases with their

length, while simultaneously removing the need for the $\lambda$ parameter. The $\gamma$-return is given by the weighting scheme:

$$w_\gamma(n, L) = \frac{(\sum_{i=1}^n \gamma^{2(i-1)})^{-1}}{\sum_{\hat{n}=1}^L (\sum_{i=1}^{\hat{n}} \gamma^{2(i-1)})^{-1}}.$$

## 3 The $\Omega$-Return

We propose a new complex return, the $\Omega$-return, that improves upon the $\lambda$ and $\gamma$ returns by accounting for the correlations of the returns. To emphasize this problem, notice that $R_{s_t}^{(20)}$ and $R_{s_t}^{(21)}$ will be almost identical (perfectly correlated) for many MDPs (particularly when $\gamma$ is small). This means that Assumption 1 is particularly egregious, and suggests that a new complex return might improve upon the $\lambda$ and $\gamma$-returns by properly accounting for the correlation of returns.

We formulate the problem of how best to combine different length returns to estimate the true expected return as a linear regression problem. This reformulation allows us to leverage the well-understood properties of linear regression algorithms. Consider a regression problem with $L$ points, $\{(x_i, y_i)\}_{i=1}^L$, where the value of $y_i$ depends on the value of $x_i$. The goal is to predict $y_i$ given $x_i$. We set $x_i = 1$ and $y_i = R_{s_t}^{(i)}$. We can then construct the design matrix (a vector in this case), $\mathbf{x} = \mathbf{1} = [1, \ldots, 1]^\mathsf{T} \in \mathbb{R}^L$ and the response vector, $\mathbf{y} = [R_{s_t}^{(1)}, R_{s_t}^{(2)}, \ldots, R_{s_t}^{(L)}]^\mathsf{T}$. We seek a regression coefficient, $\hat{\beta} \in \mathbb{R}$, such that $\mathbf{y} \approx \mathbf{x}\hat{\beta}$. This $\hat{\beta}$ will be our estimate of the true expected return.

*Generalized least squares* (GLS) is a method for selecting $\hat{\beta}$ when the $y_i$ are not necessarily independent and may have different variances. Specifically, if we use a linear model with (possibly correlated) mean-zero noise to model the data, i.e., $\mathbf{y} = \mathbf{x}\beta + \boldsymbol{\epsilon}$, where $\beta \in \mathbb{R}$ is unknown, $\boldsymbol{\epsilon}$ is a random vector, $\mathrm{E}[\boldsymbol{\epsilon}] = \mathbf{0}$, and $\mathrm{Var}(\boldsymbol{\epsilon}|\mathbf{x}) = \Omega$, then the GLS estimator

$$\hat{\beta} = (\mathbf{x}^\mathsf{T}\Omega^{-1}\mathbf{x})^{-1}\mathbf{x}^\mathsf{T}\Omega^{-1}\mathbf{y}, \qquad (2)$$

is the *best linear unbiased estimator* (BLUE) for $\beta$ [6]—the linear unbiased estimator with the lowest possible variance.

In our setting the assumptions about the true model that produced the data become that $[R_{s_t}^{(1)}, R_{s_t}^{(2)}, \ldots, R_{s_t}^{(L)}]^\mathsf{T} = [V(s_t), V(s_t), \ldots, V(s_t)]^\mathsf{T} + \boldsymbol{\epsilon}$, where $\mathrm{E}[\boldsymbol{\epsilon}] = \mathbf{0}$ (i.e., the returns are all unbiased estimates of the true expected return) and $\mathrm{Var}(\boldsymbol{\epsilon}|\mathbf{x}) = \Omega$. Since $\mathbf{x} = \mathbf{1}$ in our case, $\mathrm{Var}(\boldsymbol{\epsilon}|\mathbf{x})(i,j) = \mathrm{Cov}(R_{s_t}^{(i)} - V(s_t), R_{s_t}^{(j)} - V(s_t)) = \mathrm{Cov}(R_{s_t}^{(i)}, R_{s_t}^{(j)})$, where $\mathrm{Var}(\boldsymbol{\epsilon}|\mathbf{x})(i,j)$ denotes the element of $\mathrm{Var}(\boldsymbol{\epsilon}|\mathbf{x})$ in the $i$th row and $j$th column.

So, using only Assumption 2, GLS ((2), solved for $\hat{\beta}$) gives us the complex return:

$$\hat{\beta} = \underbrace{\left(\begin{bmatrix} 1 & 1 & \cdots & 1 \end{bmatrix} \Omega^{-1} \begin{bmatrix} 1 \\ 1 \\ \vdots \\ 1 \end{bmatrix}\right)^{-1}}_{=\frac{1}{\sum_{n,m=1}^L \Omega^{-1}(n,m)}} \underbrace{\begin{bmatrix} 1 & 1 & \cdots & 1 \end{bmatrix} \Omega^{-1} \begin{bmatrix} R_{s_t}^{(1)} \\ R_{s_t}^{(2)} \\ \vdots \\ R_{s_t}^{(L)} \end{bmatrix}}_{=\sum_{n,m=1}^L \Omega^{-1}(n,m)R_{s_t}^{(n)}},$$

which can be written in the form of (1) with weights:

$$w_\Omega(n, L) = \frac{\sum_{m=1}^L \Omega^{-1}(n, m)}{\sum_{\hat{n},m=1}^L \Omega^{-1}(\hat{n}, m)}, \qquad (3)$$

where $\Omega$ is an $L \times L$ matrix with $\Omega(i,j) = \mathrm{Cov}(R_{s_t}^{(i)}, R_{s_t}^{(j)})$.

Notice that the $\Omega$-return is a generalization of the $\lambda$ and $\gamma$ returns. The $\lambda$-return can be obtained by reintroducing the false assumption that the returns are independent and that their variance grows geometrically, i.e., by making $\Omega$ a diagonal matrix with $\Omega_{n,n} = \lambda^{-n}$. Similarly, the $\gamma$-return can be obtained by making $\Omega$ a diagonal matrix with $\Omega_{n,n} = \sum_{i=1}^n \gamma^{2(i-1)}$.

Notice that $R_{s_t}^{\Omega}$ is a BLUE of $V(s_t)$ if Assumption 2 holds. Since Assumption 2 does not hold, the $\Omega$-return is *not* an unbiased estimator of $V(s)$. Still, we expect it to outperform the $\lambda$ and $\gamma$-returns because it accounts for the correlation of $n$-step returns and they do not. However, in some cases it may perform worse because it is still based on the false assumption that all of the returns are unbiased estimators of $V(s_t)$. Furthermore, given Assumption 2, there may be *biased* estimators of $V(s_t)$ that have lower expected mean squared error than a BLUE (which must be unbiased).

# 4    Approximating the $\Omega$-Return

In practice the covariance matrix, $\Omega$, is unknown and must be approximated from data. This approach, known as *feasible generalized least squares* (FGLS), can perform worse than ordinary least squares given insufficient data to accurately estimate $\Omega$. We must therefore accurately approximate $\Omega$ from small amounts of data.

To study the accuracy of covariance matrix estimates, we estimated $\Omega$ using a large number of trajectories for four different domains: a $5 \times 5$ gridworld, a variant of the canonical mountain car domain, a real-world digital marketing problem, and a continuous control problem (DAS1), all of which are described in more detail in subsequent experiments. The covariance matrix estimates are depicted in Figures 1(a), 2(a), 3(a), and 4(a). We do not specify rows and columns in the figures because all covariance matrices and estimates thereof are symmetric. Because they were computed from a very large number of trajectories, we will treat them as ground truth.

We must estimate the $\Omega$-return when only a few trajectories are available. Figures 1(b), 2(b), 3(b), and 4(b) show direct empirical estimates of the covariance matrices using only a few trajectories. These empirical approximations are poor due to the very limited amount of data, except for the digital marketing domain, where a "few" trajectories means 10,000. The solid black entries in Figures 1(f), 2(f), 3(f), and 4(f) show the weights, $w_{\Omega}(n, L)$, on different length returns when using different estimates of $\Omega$. The noise in the direct empirical estimate of the covariance matrix using only a few trajectories leads to poor estimates of the return weights.

When approximating $\Omega$ from a small number of trajectories, we must be careful to avoid this overfitting of the available data. One way to do this is to assume a compact parametric model for $\Omega$. Below we describe a parametric model of $\Omega$ that has only four parameters, regardless of $L$ (which determines the size of $\Omega$). We use this parametric model in our experiments as a proof of concept—we show that the $\Omega$-return using even this simple estimate of $\Omega$ can produce improved results over the other existing complex returns. We do not claim that this scheme for estimating $\Omega$ is particularly principled or noteworthy.

## 4.1    Estimating Off-Diagonal Entries of $\Omega$

Notice in Figures 1(a), 2(a), 3(a), and 4(a) that for $j > i$, $\mathrm{Cov}(R_{s_t}^i, R_{s_t}^j) \approx \mathrm{Cov}(R_{s_t}^i, R_{s_t}^i) = \mathrm{Var}(R_{s_t}^i)$. This structure would mean that we can fill in $\Omega$ given its diagonal values, leaving only $L$ parameters. We now explain why this relationship is reasonable in general, and not just an artifact of our domains. We can write each entry in $\Omega$ as a recurrence relation:

$$\mathrm{Cov}[R_{s_t}^{(i)}, R_{s_t}^{(j)}] = \mathrm{Cov}[R_{s_t}^{(i)}, R_{s_t}^{(j-1)} + \gamma^{j-1}(r_{t+j} + \gamma \hat{V}(s_{t+j}) - \hat{V}(s_{t+j-1})]$$
$$= \mathrm{Cov}[R_{s_t}^{(i)}, R_{s_t}^{(j-1)}] + \gamma^{j-1}\mathrm{Cov}[R_{s_t}^{(i)}, r_{t+j} + \gamma \hat{V}(s_{t+j}) - \hat{V}(s_{t+j-1})],$$

when $i < j$. The term $r_{t+j} + \gamma \hat{V}(s_{t+j}) - \hat{V}(s_{t+j-1})$ is the temporal difference error $j$ steps in the future. The proposed assumption that $\mathrm{Cov}(R_{s_t}^i, R_{s_t}^j) = \mathrm{Var}(R_{s_t}^i)$ is equivalent to assuming that the covariance of this temporal difference error and the $i$-step return is negligible: $\gamma^{j-1}\mathrm{Cov}[R_{s_t}^{(i)}, r_{t+j} + \gamma \hat{V}(s_{t+j}) - \hat{V}(s_{t+j-1})] \approx 0$. The approximate independence of these two terms is reasonable in general due to the Markov property, which ensures that at least the conditional covariance, $\mathrm{Cov}[R_{s_t}^{(i)}, r_{t+j} + \gamma \hat{V}(s_{t+j}) - \hat{V}(s_{t+j-1})|s_t]$, is zero.

Because this relationship is not exact, the off-diagonal entries tend to grow as they get farther from the diagonal. However, especially when some trajectories are padded with absorbing states, this relationship is quite accurate when $j = L$, since the temporal difference errors at the absorbing state are all zero, and $\mathrm{Cov}[R_{s_t}^{(i)}, 0] = 0$. This results in a significant difference between $\mathrm{Cov}[R_{s_t}^{(i)}, R_{s_t}^{(L-1)}]$

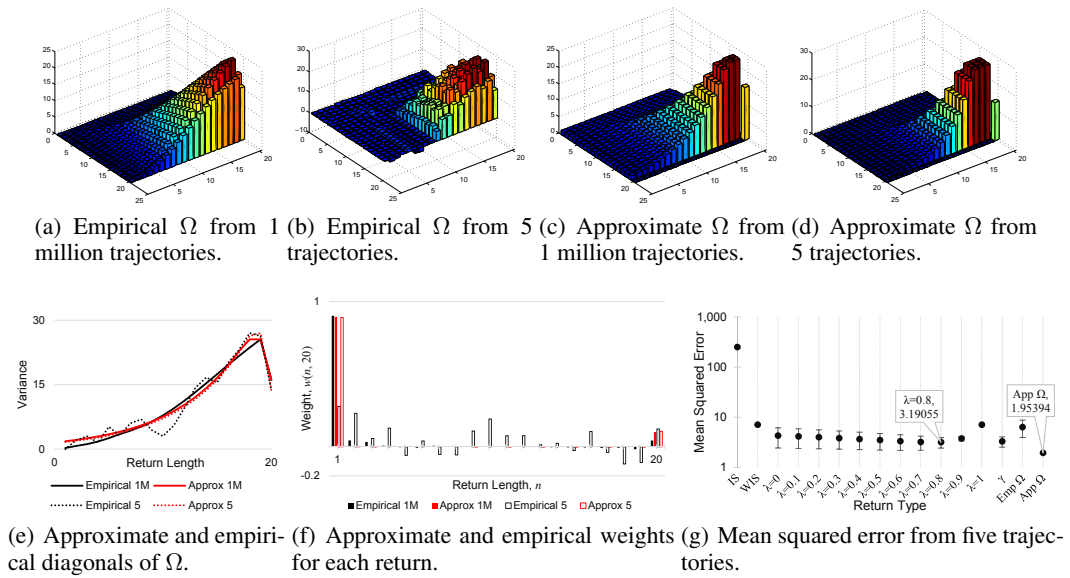

(a) Empirical $\Omega$ from 1 million trajectories.

(b) Empirical $\Omega$ from 5 trajectories.

(c) Approximate $\Omega$ from 1 million trajectories.

(d) Approximate $\Omega$ from 5 trajectories.

(e) Approximate and empirical diagonals of $\Omega$.

(f) Approximate and empirical weights for each return.

(g) Mean squared error from five trajectories.

Figure 1: Gridworld Results.

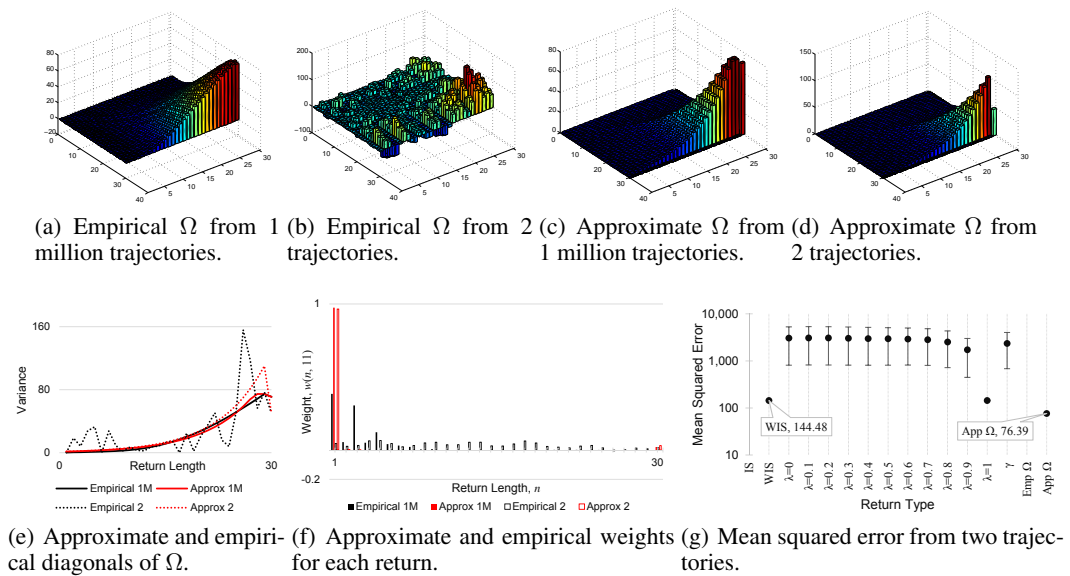

(a) Empirical $\Omega$ from 1 million trajectories.

(b) Empirical $\Omega$ from 2 trajectories.

(c) Approximate $\Omega$ from 1 million trajectories.

(d) Approximate $\Omega$ from 2 trajectories.

(e) Approximate and empirical diagonals of $\Omega$.

(f) Approximate and empirical weights for each return.

(g) Mean squared error from two trajectories.

Figure 2: Mountain Car Results.

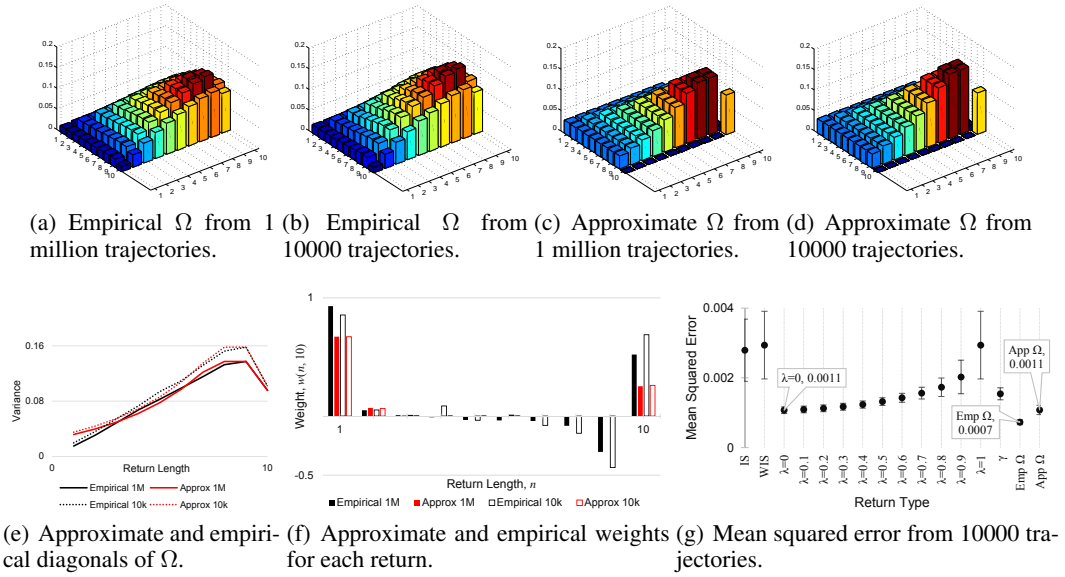

(a) Empirical $\Omega$ from 1 million trajectories.

(b) Empirical $\Omega$ from 10000 trajectories.

(c) Approximate $\Omega$ from 1 million trajectories.

(d) Approximate $\Omega$ from 10000 trajectories.

(e) Approximate and empirical diagonals of $\Omega$.

(f) Approximate and empirical weights for each return.

(g) Mean squared error from 10000 trajectories.

Figure 3: Digital Marketing Results.

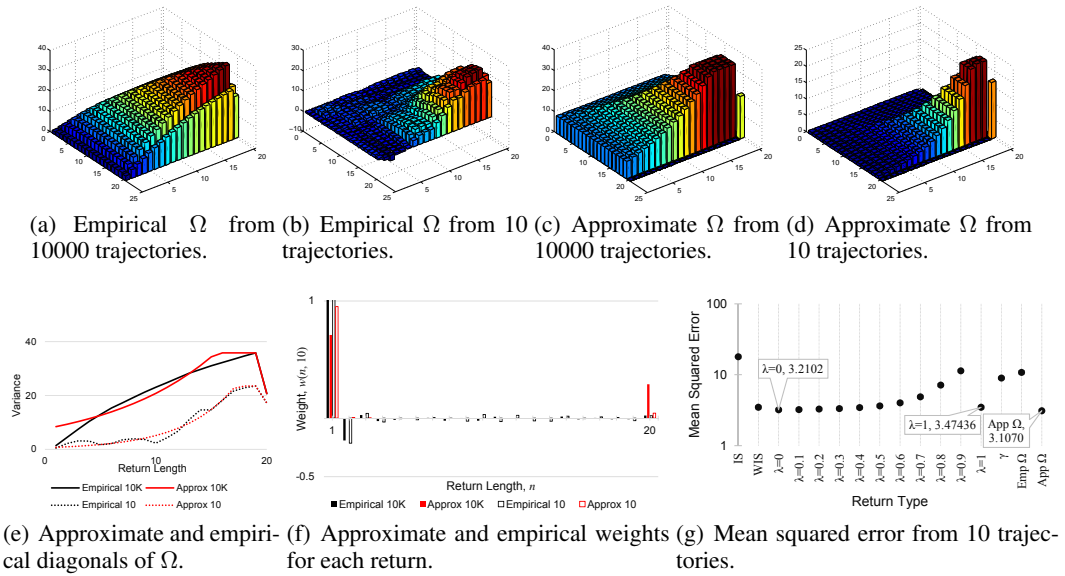

(a) Empirical $\Omega$ from 10000 trajectories.

(b) Empirical $\Omega$ from 10 trajectories.

(c) Approximate $\Omega$ from 10000 trajectories.

(d) Approximate $\Omega$ from 10 trajectories.

(e) Approximate and empirical diagonals of $\Omega$.

(f) Approximate and empirical weights for each return.

(g) Mean squared error from 10 trajectories.

Figure 4: Functional Electrical Stimulation Results.

and $\text{Cov}[R_{s_t}^{(i)}, R_{s_t}^{(L)}]$. Rather than try to model this drop, which can influence the weights significantly, we reintroduce the assumption that the Monte Carlo return is independent of the other returns, making the off-diagonal elements of the last row and column zero.

## 4.2 Estimating Diagonal Entries of $\Omega$

The remaining question is how best to approximate the diagonal of $\Omega$ from a very small number of trajectories. Consider the solid and dotted black curves in Figures 1(e), 2(e), 3(e), and 4(e), which depict the diagonals of $\Omega$ when estimated from either a large number or small number of trajectories. When using only a few trajectories, the diagonal includes fluctuations that can have significant impacts on the resulting weights. However, when using many trajectories (which we treat as giving ground truth), the diagonal tends to be relatively smooth and monotonically increasing until it plateaus (ignoring the final entry).

This suggests using a smooth parametric form to approximate the diagonal, which we do as follows. Let $v_i$ denote the sample variance of $R_{s_t}^{(i)}$ for $i = 1 \ldots L$. Let $v_+$ be the largest sample variance: $v_+ = \max_{i \in \{1,\ldots,L\}} v_i$. We parameterize the diagonal using four parameters, $k_1, k_2, v_+$, and $v_L$:

$$\hat{\Omega}_{k_1,k_2,v_+,v_L}(i,i) = \begin{cases} k_1 & \text{if } i = 1 \\ v_L & \text{if } i = L \\ \min\{v_+, k_1 k_2^{(1-t)}\} & \text{otherwise.} \end{cases}$$

$\Omega(1,1) = k_1$ sets the initial variance, and $v_L$ is the variance of the Monte Carlo return. The parameter $v_+$ enforces a ceiling on the variance of the $i$-step return, and $k_2$ captures the growth rate of the variance, much like $\lambda$. We select the $k_1$ and $k_2$ that minimize the mean squared error between $\hat{\Omega}(i,i)$ and $v_i$, and set $v_+$ and $v_L$ directly from the data.[2]

This reduces the problem of estimating $\Omega$, an $L \times L$ matrix, to estimating four numbers from return data. Consider Figures 1(c), 2(c), 3(c), and 4(c), which depict $\hat{\Omega}$ as computed from many trajectories. The differences between these estimates and the ground truth show that this parameterization is not perfect, as we cannot represent the true $\Omega$ exactly. However, the estimate is reasonable and the resulting weights (solid red) are visually similar to the ground truth weights (solid black) in Figures 1(f), 2(f), 3(f), and 4(f). We can now get accurate estimates of $\Omega$ from very few trajectories. Figures 1(d), 2(d), 3(d), and 4(d) show $\hat{\Omega}$ when computed from only a few trajectories. Note their similarity to $\hat{\Omega}$ when using a large number of trajectories, and that the resulting weights (unfilled red in Figures 1(f), 2(f), 3(f), and 4(f)) are similar to the those obtained using many more trajectories (the filled red bars).

Pseudocode for approximating the $\Omega$-return is provided in Algorithm 1. Unlike the $\lambda$-return, which can be computed from a single trajectory, the $\Omega$-return requires a set of trajectories in order to estimate $\Omega$. The pseudocode assumes that every trajectory is of length $L$, which can be achieved by padding shorter trajectories with absorbing states.

**Algorithm 1:** Computing the $\Omega$-return.

**Require:** $n$ trajectories beginning at $s$ and of length $L$.

1. Compute $R_s^{(i)}$ for $i = 1, \ldots, L$ and for each trajectory.

2. Compute the sample variances, $v_i = \text{Var}(R_s^{(i)})$, for $i = 1, \ldots, L$.

3. Set $v_+ = \max_{i \in \{1, \ldots, L\}} v_i$.

4. Search for the $k_1$ and $k_2$ that minimize the mean squared error between $v_i$ and $\hat{\Omega}_{k_1, k_2, v_+, v_L}(i, i)$ for $i = 1, \ldots, L$.

5. Fill the diagonal of the $L \times L$ matrix, $\Omega$, with $\Omega(i, i) = \hat{\Omega}_{k_1, k_2, v_+, v_L}(i, i)$, using the optimized $k_1$ and $k_2$.

6. Fill all of the other entries with $\Omega(i, j) = \Omega(i, i)$ where $j > i$. If ($i = L$ or $j = L$) and $i \neq j$ then set $\Omega(i, j) = 0$ instead.

7. Compute the weights for the returns according to (3).

8. Compute the $\Omega$-return for each trajectory according to (1).

# 5   Experiments

Approximations of the $\Omega$-return could, in principle, replace the $\lambda$-return in the whole family of TD($\lambda$) algorithms. However, using the $\Omega$-return for TD($\lambda$) raises several interesting questions that are beyond the scope of this initial work (e.g., is there a linear-time way to estimate the $\Omega$-return? Since a different $\Omega$ is needed for every state, how can the $\Omega$-return be used with function approximation where most states will never be revisited?). We therefore focus on the specific problem of *off-policy policy evaluation*—estimating the performance of a policy using trajectories generated by a possibly different policy. This problem is of interest for applications that require the evaluation of a proposed policy using historical data.

Due to space constraints, we relegate the details of our experiments to the appendix in the supplemental documents. However, the results of the experiments are clear—Figures 1(g), 2(g), 3(g), and 4(g) show the *mean squared error* (MSE) of value estimates when using various methods.[3] Notice that, for all domains, using the $\Omega$-return (the EMP $\Omega$ and APP $\Omega$ labels) results in lower MSE than the $\gamma$-return and the $\lambda$-return with any setting of $\lambda$.

# 6   Conclusions

Recent work has begun to explore the statistical basis of complex estimates of return, and how we might reformulate them to be more statistically efficient [4]. We have proposed a return estimator that improves upon the $\lambda$ and $\gamma$-returns by accounting for the covariance of return estimates. Our results show that understanding and exploiting the fact that in control settings—unlike in standard supervised learning—observed samples are typically neither independent nor identically distributed, can substantially improve data efficiency in an algorithm of significant practical importance.

Many (largely positive) theoretical properties of the $\lambda$-return and TD($\lambda$) have been discovered over the past few decades. This line of research into other complex returns is still in its infancy, and so there are many open questions. For example, can the $\Omega$-return be improved upon by removing Assumption 2 or by keeping Assumption 2 but using a biased estimator (not a BLUE)? Is there a method for approximating the $\Omega$-return that allows for value function approximation with the same time complexity as TD($\lambda$), or which better leverages our knowledge that the environment is Markovian? Would TD($\lambda$) using the $\Omega$-return be convergent in the same settings as TD($\lambda$)? While we hope to answer these questions in future work, it is also our hope that this work will inspire other researchers to revisit the problem of constructing a statistically principled complex return.

## Footnotes

[1]To be clear: there *is* a wealth of theoretical and empirical analyses of algorithms that use the $\lambda$-return. Until recently there was *not* a derivation of the $\lambda$-return as the estimator of $V(s_t)$ that optimizes some objective (e.g., maximizes log likelihood or minimizes expected squared error).

[2]We include the constraints that $k_2 \in [0, 1]$ and $0 \le k_1 \le v_+$.

[3]To compute the MSE we used a large number of Monte Carlo rollouts to estimate the true value of each policy.

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
