[Supplementary Material · Appendix.pdf]

## Appendix 1: Off-Policy Evaluation

The performance of a policy is the expected return of a trajectory generated when following that policy, i.e., the value of the initial state.[4] Given a set $D$ of $m$ trajectories, $\tau_i$, $i = 1, \ldots, m$, generated by a *behavior policy*, $\pi_b$, and an *evaluation policy*, $\pi_e$, we must estimate the expected return of $\pi_e$: $V^{\pi_e}(s_0) = \mathrm{E}[R_{s_0}^{MC}|\pi_e]$. We now describe existing methods for off-policy evaluation and then propose a new approach that uses the $\Omega$-return.

### Importance Sampling

If the trajectories were on-policy (i.e., $\pi_b = \pi_e$) we could estimate the expected return of $\pi_e$ by averaging the observed returns. However, when the evaluation policy differs from the behavior policy, we must take a weighted average of the returns where larger weights are given to trajectories that are more likely under the evaluation policy. *Importance sampling* (IS), a principled way to perform this reweighting, produces an unbiased estimate of $V^{\pi_e}(s_0)$ from each trajectory as:

$$\hat{V}_{\mathrm{MC}}^{\pi_e}(s_0, \tau) = R_{s_0}^{MC} \underbrace{\prod_{t=0}^{L-1} \frac{\pi_e(s_t^\tau, a_t^\tau)}{\pi_b(s_t^\tau, a_t^\tau)}}_{\text{Importance Weight}},$$

where $s_t^\tau$ and $a_t^\tau$ are the state and action at time $t$ in trajectory $\tau$, $L$ is the length of the longest trajectory,[5] and $\pi(s, a)$ denotes the probability of action $a$ in state $s$ under policy $\pi$. $\hat{V}_{\mathrm{MC}}^{\pi_e}(s_0, \tau_i)$ is called the *importance weighted return*. The estimate of $V^{\pi_e}(s_0)$ from $D$ can then be produced by averaging the importance weighted return from each trajectory:

$$\hat{V}_{\mathrm{IS}}^{\pi_e}(s_0, D) = \frac{1}{m} \sum_{\tau \in D} \hat{V}_{\mathrm{MC}}^{\pi_e}(s_0, \tau).$$

Although $\hat{V}_{\mathrm{IS}}^{\pi_e}(s_0, D)$ is an unbiased estimate of $V^{\pi_e}(s_0)$, it has high variance since the importance weights can be large for long trajectories [7].

### Weighted Importance Sampling

In *weighted importance sampling* (WIS) the sum of the importance weighted returns is divided by the sum of importance weights rather than the number of trajectories:

$$\hat{V}_{\mathrm{WIS}}^{\pi_e}(s_0, D) = \frac{1}{\sum_{\tau \in D} \prod_{t=0}^{L-1} \frac{\pi_e(s_t^\tau, a_t^\tau)}{\pi_b(s_t^\tau, a_t^\tau)}} \sum_{i=1}^{n} \hat{V}_{\mathrm{MC}}^{\pi_e}(s_0, \tau_i).$$

This modification means that $\hat{V}_{\mathrm{WIS}}^{\pi_e}(s_0, D)$ is a *biased* but consistent estimator of $V^{\pi_e}(s_0)$. It also tends to have significantly lower variance than $\hat{V}_{\mathrm{IS}}^{\pi_e}(s_0, D)$; WIS trades bias for lower variance relative to IS, and is the current method of choice for off-policy evaluation [7, 8].

### Complex Weighted Importance Sampling

We propose a simple modification to WIS, which we call *Complex Weighted Importance Sampling* (CWIS). Whereas IS and WIS use the Monte Carlo return in $\hat{V}_{\mathrm{MC}}^{\pi_e}(s_0, \tau)$, CWIS uses a complex return like $R_{s_0}^\lambda$, $R_{s_0}^\gamma$, or $R_{s_0}^\Omega$. This replaces the unbiased Monte Carlo estimate of return with a lower variance but biased (and not necessarily consistent) estimator of expected return. This is a worthwhile trade-off because the primary drawback of IS and WIS is their high variance. We choose to focus on using complex returns for WIS rather than IS because the use of complex backups would make IS biased, damaging its primary selling point.

The CWIS$_\dagger$ estimator of $V^{\pi_e}(s_0)$ is:

$$\hat{V}_{\mathrm{CWIS}_\dagger}^{\pi_e}(s_0, D) = \frac{1}{\sum_{\tau \in D} \prod_{t=0}^{L-1} \frac{\pi_e(s_t^\tau, a_t^\tau)}{\pi_b(s_t^\tau, a_t^\tau)}} \sum_{i=1}^{n} \hat{V}_\dagger^{\pi_e}(s_0, \tau_i), \tag{4}$$

where $\hat{V}_\dagger^{\pi_e}(s_0, \tau) = R_{s_0}^\dagger \prod_{t=0}^{L-1} \frac{\pi_e(s_t^\tau, a_t^\tau)}{\pi_b(s_t^\tau, a_t^\tau)}$, and $\dagger$ specifies the complex return, e.g., $\lambda$, $\gamma$, $\Omega$, or $0$. Notice that CWIS$_{\mathrm{MC}}$ is equivalent to WIS and CWIS$_{\lambda=1}$. Also, CWIS$_{\lambda=0}$ is the ECR estimator [9].

CWIS assumes that an estimate of the off-policy value function is available when computing returns of different lengths. Any off-policy evaluation method can be used to produce an estimate of the value function for $\pi_e$. Ideally this off-policy value function approximation would be computed to minimize the mean squared error between the $\Omega$-return and the approximation of the value function. However, value function approximation methods for the $\Omega$-return are not yet available, so in their absence we use *WIS-LSTD*$(0.5)$ [8] with the Fourier basis [10].

An implementation of CWIS$_\dagger$ has four steps. **1)** Estimate the off-policy value function. Here we used WIS-LSTD$(0.5)$. **2)** Compute the different length returns $R_{s_0}^{(n)}$ for each trajectory for $n = 1, \ldots, L$. **3)** Compute the $\dagger$-return, $R_{s_0}^\dagger$, for each trajectory. **4)** Use (4) to compute the final estimate of the performance of $\pi_e$.

Note that if the complex return used by CWIS$_\dagger$ is the same as that used by the off-policy value function approximation method, then CWIS$_\dagger$ returns the *target* of the off-policy value function approximation method for start state $s_0$.

# Appendix 2: Off-Policy Evaluation Experiments

We compared IS, WIS, CWIS$_0$, CWIS$_\lambda$ (for various $\lambda$), CWIS$_\gamma$, and CWIS$_\Omega$ on four benchmark domains. In three cases CWIS$_\Omega$ using our approximation to $\Omega$ outperforms all other methods, and in the fourth case it performs as well as every other method tested except for CWIS$_\Omega$ using a direct empirical estimate of $\Omega$.

$5 \times 5$ **Gridworld.** Our first empirical study used a $5 \times 5$ gridworld [5] with a reasonable hand-tuned behavior policy and an improved evaluation policy. We sampled 1 million trajectories from the evaluation policy and used the average return as the ground truth value of the evaluation policy. We then repeated the following process 10000 times: sample 5 trajectories and use IS, WIS, CWIS$_0$, CWIS$_\lambda$ (with 11 different values of $\lambda$), CWIS$_\gamma$, and CWIS$_\Omega$ to estimate the value of the evaluation policy. The *mean squared error* (MSE) over these 10000 trials, including standard error bars, is shown in Figure 1(g). In this and subsequent similar figures, "Emp $\Omega$" (empirical) denotes CWIS$_\Omega$ using the sample covariance matrix to estimate $\Omega$ while "App $\Omega$" (approximate) denotes CWIS$_\Omega$ using our proposed scheme for smoothly approximating $\Omega$. Notice that our scheme for approximating $\Omega$ results in CWIS$_\Omega$ having the lowest MSE, with CWIS$_\lambda$ with $\lambda = 0.8$ having the second lowest MSE.

**Mountain Car.** Our second study was set up similarly, using a simplified variant of the canonical mountain car domain [5], where each action was held constant for twenty time steps to shorten trajectories and avoid numerical overflows with IS. The behavior policy was random, the evaluation policy near-optimal, and only two trajectories were used for each estimate of the evaluation policy's performance (this makes the problem challenging for CWIS, since the estimate of the value function is poor). The MSEs from this experiment are depicted in Figure 2(g).

Notice the large MSE, which are due to the disparity between the behavior and evaluation policies. This caused large importance weights which can cause instability in value function approximation methods like WIS-LSTD when using only two trajectories to estimate the value function. As a result, the accuracy of the approximate value function produced by WIS-LSTD can vary. CWIS$_{\lambda=1}$, which is equivalent to WIS, therefore performs well because it does not use the approximate value function produced by WIS-LSTD. However, "Approx $\Omega$" performs best, selectively using the value function only when it is accurate enough to produce low variance returns.

**Digital Marketing.** Our third case study involved targeting advertisements. When a user visits a webpage, a vector of features describing the user is used to select an advertisement to display. A well-targeted advertisement will be of interest to the user and has a higher probability of being clicked than a poorly-targeted one. It has been shown that the problem of which advertisement to show should be treated as a sequential decision problem (as opposed to a bandit problem) [11].

We used the same digital marketing simulator as Thomas et al. [12], which was trained using real data from a Fortune 50 company, to evaluate a newly optimized policy using 10000 trajectories from a previous (slightly worse) policy. This process was repeated 100 times to produce Figure 3(g). Notice that here we are using significantly more trajectories to form each estimate of the evaluation policy's performance. This is because the simulator was built using real data from a company that receives hundreds of thousands of unique visitors per day, so that 10000 trajectories is a "small" amount of data. With so many trajectories, we can use the sample covariance matrix to accurately estimate $\Omega$ for CWIS$_\Omega$, and hence this approach ("Emp $\Omega$") performs best, though our approximate $\Omega$ performs as well as the best setting of $\lambda$.

**Functional Electrical Stimulation.** Our fourth study used a simulation, *Dynamic Arm Simulator 1* [13], of a human arm undergoing *functional electrical stimulation* (FES). The goal of FES is to use direct stimulation of the muscles in a paralyzed subject's arm to move the arm from the current position to a desired position. Researchers have used RL previously for simulated studies into the efficacy of RL for automatically optimiz-

ing the controller for each individual's arm and to mitigate sources of nonstationarity including fatigue [14]. Researchers have also considered using RL for control of ordinary prosthetics [15].

Here we use the same arm model and domain specifications as Thomas et al. [14], except with longer time steps in order to decrease trajectory lengths to avoid numerical instabilities in IS. The behavior policy performs similar to a *proportional derivative* (PD) controller optimized for FES control of a human arm [16], and the evaluation policy is a slightly improved policy found using the CMA-ES policy search algorithm [17].

In a real deployment of FES, each trajectory is a two-second reaching movement, and so a controller that can adapt using only tens of trajectories is important. We therefore consider the problem of evaluating a proposed policy improvement using only ten trajectories. This is repeated 1000 times, and the resulting MSEs are provided in Figure 4(g). $\text{CWIS}_\Omega$ using our approximation performs best.

## Footnotes

[4]A distribution over initial states can be handled by inserting a new initial state with only one admissible action, no reward, which transitions according to the start distribution.

[5]All shorter trajectories are padded to be of length $L$ using a zero-reward absorbing state with one admissible action.