[Reviews · NeurIPS 2015]

Submitted by Assigned_Reviewer_1

The paper presents a clear and readable comparison of classical lambda-returns (that lie at the heart of TD(lambda)), gamma-returns (which were introduced more recently) and Omega-returns (which are derived in the current paper). I think this perspective (which builds on the insights in the gamma-return work) is really interesting and instructive. I don't think the new algorithm that was presented is all that helpful, but the paper provides valuable insights on an important and complex core problem in learning.

Quality: very good Clarity: excellent Originality: good Significance: good

Line 068: Can L be infinity? If not, can you be more specific about how these results relate to standard applications with no fixed step length?

Line 078: I think L is missing from w_lambda(n).

Line 270: I think the figures would be easier to compare if the axes were fixed between plots. I think I'm supposed to be relating (a) to (d) and it's difficult to do so with the varying scales.

Line 282: So, it looks like many of the (f) plots show a pattern that involves a lot of weight on the one-step estimator (TD(0)) and sometimes a bit on TD(1). Does this work provide additional insight into how good estimator TD(0) is? It seems like it's quite easy to compute and perhaps nearly as good as what is coming out of the complex estimator. Some insight into what is gained by the small weights on the other estimators would be illuminating.

Summary: The paper presents a clear and readable comparison of classical lambda-returns (that lie at the heart of TD(lambda)), gamma-returns (which were introduced more recently) and Omega-returns (which are derived in the current paper). I think this perspective (which builds on the insights in the gamma-return work) is really interesting and instructive. I don't think the new algorithm that was presented is all that helpful, but the paper provides valuable insights on an important and complex core problem in learning.

Submitted by Assigned_Reviewer_2

***After author response/discussion***

I have moved my score up for two main reasons: 1) The authors helpful response did clarify the experimental set-up for me, which I agree does not need to include True TD(lambda). I have edited my review below to reflect this.

2) I have come around to thinking of the approximate algorithm as a proof of concept rather than a primary contribution. Seeing this paper as describing a solid foothold in this direction, and some early evidence that there may be something to be gained by going further makes me less concerned about its practical impact, and more optimistic about its chances of provoking questions and possibly inspiring follow-on work.

---Paper Summary---

This paper proposes a novel replacement for the lambda return for mixing between bootstrapping and Monte Carlo estimates in reinforcement learning. Specifically, the Omega return in principle accounts for the covariance of returns at different depths (whereas the lambda return implicitly assumes the returns at various depths are independent). Computation of the Omega return is formulated as a generalized least squares problem, which impractically relies upon unavailable quantities. The paper describes one method for approximating these quantities using experience data. Empirical results in four domains suggest that the approximation produces qualitatively sensible weights, and that the use of the approximate Omega return yields more accurate off-policy estimates of policy quality.

---Quality---

I believe the derivation of the Omega return weights (the main technical result) is technically sound. The various steps of approximation are generally well-motivated. One argument that I did not totally follow starts on line 208. The argument is intended to motivate the assumption that the covariance between two different depth returns is approximately equal to the variance of the shallower return. Intuitively I can see why this might be reasonable; it seems sensible in many problems to expect the return at one depth to be very similar to a slightly deeper return (less so for much deeper returns, but therein lies the approximation). The thing I didn't get is the argument that the i-step return should be nearly independent of the depth-j TD-error. The claim seems to be that they should be nearly independent because they are *conditionally* independent given state. As I'm sure the authors are aware, there's no reason that two conditionally independent variables should also be independent. If you want to assume those two quantities are independent, it's important to develop a better founded sense of what that assumption really means.

The claim that the approximation yields accurate weights is a little imprecise. It is mainly supported by eyeballing the approximated quantities to see that they are qualitatively similar to the targets. I'd have preferred a more quantitative study of the accuracy of the approximation, perhaps studying the relative impact of the various stages of approximation proposed in Section 4.

The empirical claim that using the approximate Omega return yields more accurate value estimates than the usual lambda return seems well-supported.

---Clarity---

I found the paper to be well-written for the most part. I particularly appreciated the introduction which I thought laid out the issues at hand clearly and effectively set the stage for the technical contributions. The main area for improvement in clarity I see is Section 4. The section takes the reader through multiple different stages of approximation and it starts to feel a bit like a run-on sentence ("and then...and then..."). I'd humbly suggest that the authors take another look at this section with eye toward providing the reader with more structure to hold on to. Some things that might help include additional subheadings, some text along the way to place the current step in the overall approximation plan, and/or perhaps an executive summary at the end of the various assumptions and their consequences.

The other major impediment to clarity is the relegation of the description of the experiments to the supplementary material. This make it harder to determine the meaning of the results; the figures even contain terms that are not defined anywhere in the main text. The authors should be aware that they are permitted to place their references section on the 9th page, leaving nearly 3/4 page of available space. While I understand that the appendix as-is cannot fit in that space, I expect that with some judicious editing significantly more detail regarding the experiments can fit in the main text.

---Originality---

As far as I am aware, the Omega return is a novel concept, and the derivation and subsequent approximation presented here are also original, though they make use of well-known machine learning tools.

---Significance---

The conceptual problem with the lambda return is very real, and effectively addressing the bias in the estimator could potentially have an important impact on RL practice. However, this is not a novel insight -- the issues with the lambda return have already been discussed in the literature. So the burden of significance falls to the Omega return itself.

The principle underlying the Omega return is a good one, and I wouldn't say that its formulation was a priori obvious. So one source of potential significance (that the paper itself raises) is that the Omega return, as described here, may now serve as a target for approximation for future study.

I do see potential value in having this idea available in the literature as a foundation for future work.

The other possible source of significance is the particular approximation described here. Unfortunately, to my eyes it looks like this method is fairly complicated and limited compared to the lambda return. As George Box is famous for saying "All models are wrong, but some are useful." Both the approximate Omega return and lambda return are wrong, but from these results I'd be forced to conclude that the lambda return is currently more useful. I doubt the specific approximation method in this paper will significantly impact RL practice. At best it serves as a proof of concept that one can approximate the Omega return to some extent, giving hope that a better approximation will come down the road eventually.
Summary: The motivation for this paper is of genuine interest, though the practical impact of the presented approach seems small in exchange for a great deal of complication. In the end, I am convinced that the clear statement of the problem of approximating/improving upon the Omega return may be of sufficient significance.

Submitted by Assigned_Reviewer_3

The paper deals with policy evaluation (for a single state) and proposes an alternative to eligibility traces for weighting n-step returns to provide a value estimate. It starts by presenting complex returns (schemes for weighting n-step returns, typically eligibility traces, but also the more recent gamma-return). Then, the authors formalize the problem of finding the weights as a generalized-least squares problem, such as minimizing the variance. The necessary covariance matrix (Omega) being unknown, it is estimated from data. To avoid overfitting, an approximation is done (replacing the covariance between two returns by the variance of the shortest return, this variance being itself parameterized). Then, some experiments in an off-policy setting (not addressed in the paper) are provided.

Major comments: * the addressed topic is important, but this work is too preliminary. The proposed approach is an approximation of an ideal algorithm based on an unrealistic assumption. Moreover, it is hardly applicable (as it requires a set of independent trajectories and a value function estimate over the whole state space to estimate a single value). * the whole paper deals with on-policy learning, this setting should have been considered in the experimental section (with possibly additional experiments in the off-policy setting). Also, it would have been better to study bias and variance of the estimates (even if the RMSE is related to these quantities) * the proposed approach could have been compared to the (recent) emphatic TD learning approach of Sutton et al. (especially regarding the off-policy aspect)

Minor comments: * overall, the writing could be improved (notably better introduction of problem and notations, more formal writing---not use var for the variance and the empirical variance for example---, etc) * l.87: this equation makes no sense * l.210 and 213: index t+j for the reward -> t+j-1 * in the 5x5 gridworld, the true value function (ground truth) could be easily computed
Summary: This paper proposes an alternative to eligibity traces for weighting n-step returns to provide a value estimate of a given state. The problem addressed here is important and interesting, but the writing could be improved and especially this work is too preliminary (acknowledging that this is an interesting research direction): the proposed approach is an approximation of an ideal algorithm based on unreasonable assumptions and is only applicable in very constrained conditions, and the experiments do not address the first motivation of the paper (namely, improving the bias-variance tradeoff of eligibility traces in the on-policy case).

Submitted by Assigned_Reviewer_4

130: "...the unbiased estimator with the lowest..." -- that is a *linear* function of y, don't forget. 157: "Notice that ... if Assumption 2 holds. Stil..." Make a better argument why this might still be reasonable even if your model is misspecified. -The issue of correlated return estimates comes up in the GPTD and related methods, I believe. May be worth looking for connections. -Constraints on estimated \Omega probably invalidate consistency/asymptotic unbiasedness results. Comment?
Summary: The authors formulate a weighted-return estimate of value using GLS that similar in concept to TD-\lambda. Core idea is good; not clear how practical it is and what the consequences of approximation approaches are.

Author Feedback
Author rebuttal: Thank you for your feedback. We will try to address the major questions:

Reviewer 2:

> I believe the experiments are missing an important comparison

If the important comparison is true online algorithms:

Our primary goal was to compare the lambda and omega returns, not the algorithmic machinery that can be built around them. Our comparisons are *not* to TD(lambda) (true online or not). In our experiments we fix the algorithmic machinery and vary only the complex return used. That is, we compare the lambda-return to the omega-return while fixing all other aspects of the algorithm. Also, the lambda-return is computed from batch data generated by a fixed policy using the *forward* view. So, the lambda-return that we use is the desired target that the true online methods strive to produce.

If the important comparison is emphatic algorithms:

The emphatic algorithms also make use of complex returns. The appropriate comparison would be emphatic TD using the lambda-return and emphatic TD using the omega-return. One might be able to view the emphasis at time t as a modification to the weight, lambda^t, given to the update at time t, to produce a new complex return. However, we are aware of no papers that make this argument. Still, in the setting used in our experiments (evaluating only the initial state), this new complex return degenerates to a lambda-return (since the only state with non-zero emphasis is the initial state).

Furthermore, our experiments are for off-policy *policy* evaluation. That is, we only care about the value of the initial state. The GTD family of algorithms is for off-policy evaluation (for some distribution over states). When used only with an emphasis on the initial state, they degenerate to simple importance sampling (which we compare to).

> I'd be forced to conclude that the lambda return is currently more useful.
We agree - the omega return and its approximation are more complicated and more challenging to compute. For many (most?) applications (e.g., if the per-time step complexity must be linear in the number of features) the lambda-return is the clearly superior choice. However, we do not feel that this is reason to abandon this line of research:

1) Approximations of the omega-return, even in their primitive state, are already of practical use in some cases, including off-policy policy evaluation (as shown in our experiments).

2) The lambda-return has decades of research improving it (including the recent true online work). It is our hope that future similar advancements might allow for efficient approximation of the omega return.

3) We view the lambda-return as an efficient but poor approximation of the omega-return. There is an abundance of possible approximations to consider, ranging from computationally-efficient but poor approximations to inefficient but good approximations. For another efficient approximation, we could use the lambda-return but where lambda is automatically set to approximate the omega-return as closely as possible. In this paper we are proposing the omega-return, and so a detailed comparison of all possible approximation schemes is beyond the scope of this paper. We look forward to other researchers each proposing their own approximation that suits their personal desires, be they linear time complexity at the expense of performance, or maximizing performance at the expense of computational complexity.

Reviewer 6:

> Comment?
You are correct - the approximate omega-return is not a BLUE or even necessarily an unbiased estimator given the assumptions under which the omega-return is a BLUE.

ALL REVIEWERS:
The lambda-return is at the foundation of RL, but is rarely questioned. This paper attempts to change this - we question the motivation behind the lambda-return and find that it is not the "best" possible target. The primary goal of this paper is not to provide a single specific TD algorithm with a new complex return, but to study what the target for TD algorithms should ideally be. In short, this paper is about making progress in our understanding of complex returns - it is not about providing a specific algorithm that can immediately replace and improve upon the entire family of TD(lambda) algorithms.